# Uncovering the complex relationship between balding, testosterone and skin cancers in men

Jue-Sheng Ong [1] ✉, Mathias Seviiri[1,2], Jean Claude Dusingize[1], Yeda Wu [1], Xikun Han [1,3], Jianxin Shi [4], Catherine M. Olsen [1,5], Rachel E. Neale[1], John F. Thompson [6,7], Robyn P. M. Saw[6,7], Kerwin F. Shannon [6,7], Graham J. Mann [6,8,9], Nicholas G. Martin [10], Sarah E. Medland [10], Scott D. Gordon [10], Richard A. Scolyer [6,8,11,12], Georgina V. Long [6,8,13], Mark M. Iles [14,15], Maria Teresa Landi [4], David C. Whiteman [1], Stuart MacGregor [1] & Matthew H. Law [1,2] ✉

Male-pattern baldness (MPB) is related to dysregulation of androgens such as testosterone. A previously observed relationship between MPB and skin cancer may be due to greater exposure to ultraviolet radiation or indicate a role for androgenic pathways in the pathogenesis of skin cancers. We dissected this relationship via Mendelian randomization (MR) analyses, using genetic data from recent male-only meta-analyses of cutaneous melanoma (12,232 cases; 20,566 controls) and keratinocyte cancers (KCs) (up to 17,512 cases; >100,000 controls), followed by stratified MR analysis by body-sites. We found strong associations between MPB and the risk of KC, but not with androgens, and multivariable models revealed that this relationship was heavily confounded by MPB single nucleotide polymorphisms involved in pigmentation pathways. Site-stratified MR analyses revealed strong associations between MPB with head and neck squamous cell carcinoma and melanoma, suggesting that sun exposure on the scalp, rather than androgens, is the main driver. Men with less hair covering likely explains, at least in part, the higher incidence of melanoma in men residing in countries with high ambient UV.

The most common three types of skin cancers are named for their presumed cell of origin: the keratinocyte derived cancers (KC) basal cell carcinoma (BCC) and squamous cell carcinoma (SCC), and the melanocyte-derived melanoma. Exposure to ultraviolet radiation (UVR) is a key risk factor for skin cancer, and cutaneous melanocytes produce and distribute protective pigment-containing melanosomes to surrounding skin keratinocyte cells to protect against UV damage [For review see ref. 1]. In some Western countries such as Australia and the United States, cutaneous melanoma incidence is higher in men compared to women[2–5]. While this may in part derive from differences in sun exposure behaviours and reduced healthcare engagement

amongst men, these are insufficient to fully explain the difference[2]. Differences between men and women in immune responses and hormone levels may also influence the risk of melanoma[2] and KCs[6].

Male-pattern baldness (MPB), also known as androgenetic alopecia, is characterised by progressive hair follicle miniaturisation, leading to hair loss in men. This condition is often linked with changes in dihydrotestosterone levels, with several studies showing that patients with MPB had higher endogenous testosterone levels than controls[7–9]. Recent genetic studies[10–12] of endogenous testosterone levels revealed genes that are associated with both testosterone levels and MPB. MPB has been associated with a greatly increased risk of scalp melanoma

---

(hazard ratio [HR] = 7.2, 95% confidence interval [CI]: 1.3–39.4) and SCC (HR = 7.1, 95% CI: 3.8–13.1)[13]. This is of interest as the numbers of head and neck melanomas have increased by over 50% between 1994 and 2015, primarily in males of European descent in the United States and Canada[14]. Melanomas of the head and neck, and particularly the scalp, are associated with higher mortality than other sites[15,16]. Across all studied populations, men experience higher rates of head and neck melanoma[5]. Cutaneous melanoma proliferation may be influenced by androgens either directly or through immune system suppression[2,17,18], providing a potential link between MPB and cutaneous melanoma risk. A recent observational study reported a positive association between free testosterone and melanoma risk[19], but whether this relationship is potentially mediated through MPB remains unknown. For KCs, a previous Canadian study revealed that the incidence of KC is higher in men[20]. Another retrospective cohort study from Australia showed potential sex differences in the incidence of KC in different body sites, where men are more likely to develop SCC in the scalp region[21]. To the best of our knowledge, there have been no large-scale studies to date evaluating the link between testosterone and KCs to dissect sex differences in KC incidences in these populations.

The increased incidence of skin cancers amongst men with MPB may: a) reflect higher chronic UV damage to the exposed scalp; b) result from a direct causal role for male-specific factors, including testosterone (consistent with the proposed androgen basis of melanoma hypothesis[13]); or c) be driven by other factors related to androgenic regulation of immune response. Leveraging the availability of large-scale genetic data Mendelian randomization (MR)[22], a powerful genetic-based instrumental variable technique relying on the use of strong genetic proxies, can be used to detect and dissect causal pathways between MPB and skin cancers. As the allocation of genetic variants (namely single nucleotide polymorphisms, SNPs) is randomized at meiosis, the measured causal effect is less likely to be biased by reverse causality and confounding factors[23]. When used in combination with multivariable techniques[24], we can extend MR frameworks to disentangle the complex (and potentially mediated) relationship between endogenous testosterone, MPB and the risk of skin cancers.

In this study, we examined the link between endogenous testosterone levels, MPB, and skin cancers through a series of univariable and multivariable MR (MVMR) analyses. Using skin cancer outcome data from the UK Biobank (UKB)[25], the QSkin Sun and Health Study (QSkin)[26], and a recent large-scale cutaneous melanoma meta-analysis[27], we comprehensively assessed the relationship between genetic instruments for MPB, sex hormones, and skin cancer susceptibility, followed by an exploratory analysis of whether these associations differed by anatomical location (body site) of the primary cancer site.

## Results

### Methodology overview

In brief, we adopted a two-sample MR framework[22] to investigate the relationship between testosterone, MPB and risk of KCs (all KC, SCC only and BCC only) and melanoma. The skin cancer genome-wide association studies (GWASs) were derived from a recent melanoma GWAS meta-analysis[27] (participating studies in Supplementary Table 1) and the Australian QSkin cohort[26] and UKB[25] for KCs. GWAS analyses of risk factors (sex hormones) were conducted in subsets of UKB independent of those used in the melanoma GWAS meta-analysis to increase the robustness of MR findings. We first applied standard univariable MR techniques to examine the direct association between hormonal risk factors, MPB and skin cancers, followed by a multivariable approach to fit all risk factors simultaneously whilst including proxy traits captured by heterogeneous genetic outliers (see Fig. 1 for the complete study schematic). We finally evaluated site-specific cancers in a subset of cases from UKB and Melanoma Institute Australia (MIA) cohorts to assess whether these MR associations differed by primary tumour anatomic site (i.e. head and neck) (distribution of cases by anatomic site provided in Supplementary Tables 2–4; classifications in Supplementary Fig. 1), as reported in previous observational studies[21].

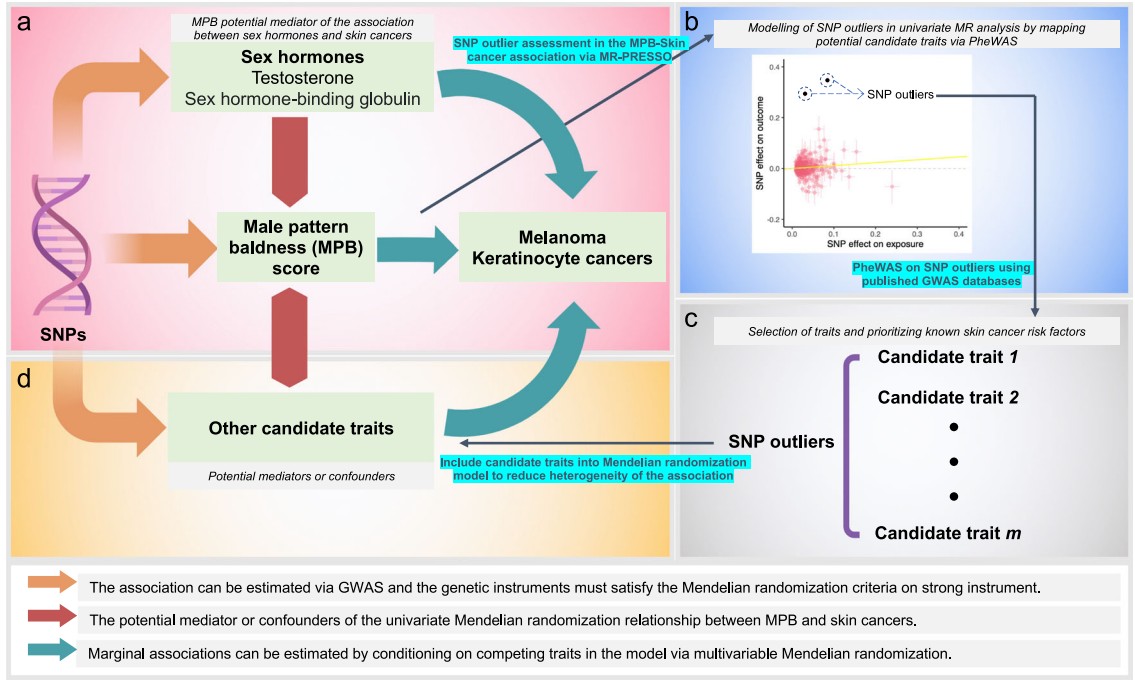

**Fig. 1 | Schematic diagram outlining the overall study approach of modelling genetic outliers via MVMR.** Each panel (**a**), (**b**), (**c**) and (**d**) is listed in chronological order of the analysis procedure. **a** Schematic MR diagram. **b** MR scatter plot. **c** Selection of candidate traits for inclusion into MVMR via PheWAS findings. **d** Modelling the candidate traits into the MVMR analysis to obtain the marginal effect of MPB on skin cancer risk, by conditioning on endogenous testosterone levels and other candidate traits.

## Assessment of statistical power for Mendelian randomization

We identified a total of 444 SNP instruments for male-pattern baldness explaining 13.95% of the phenotypic variance. Similarly, 118, 83 and 204 SNPs were used as genetic instruments for endogenous totalT, (estimated) freeT and SHBG in the univariable MR analyses, with these SNPs cumulatively explaining, 8.13, 4.94 and 14.5% of the phenotypic variances on these traits respectively. The comparison of our testosterone SNP associations (estimated in the non-overlapping samples) with those obtained in the testosterone GWAS reported in Ruth et al.[11] is shown in Supplementary Figs. 2–4, revealing highly concordant genetic effect sizes. Even when we restrict the analysis to SNPs with an association of $p < 1 \times 10^{-5}$ with the exposure of interest within our independent UKB subset (halved the original discovery GWAS sample size), the cumulative variances explained by SNPs were largely unaffected (Table 1). With the relatively high proportion of variance tagged by our SNPs, our MR study has very good power (at least 90%) to detect associations at OR > 1.2 for a one SD change in the aforementioned risk factors (Supplementary Fig. 5). The conditional F-statistics (to quantify instrument strength of our combined instrument) in the MVMR setting for each tested combination of models is shown in Supplementary Table 5.

## Primary MR analyses evaluating the MPB association with risk of skin cancers

For melanoma, none of the risk factors were associated with the risk of melanoma, with OR point estimates close to one (estimated IVW OR between 0.96 to 0.99) (Table 2). For KCs, the estimated OR per SD increase in MPB score was 1.17 (95% confidence interval [CI] 1.08–1.27) for risk of all KC, 1.15 (95% CI 1.05–1.26) for BCC and 1.31 (95% CI 1.17–1.46) for SCC in the UKB. Estimates from QSkin yielded slightly smaller effect sizes, albeit with largely overlapping confidence intervals (Table 2). The fixed-effect meta-analysed MR estimates for MPB on KC outcomes across both cohorts were: KC 1.15 (95% CI 1.06–1.23), BCC 1.15 (95% CI 1.06–1.25), and SCC 1.28 (95% CI 1.15–1.43). We detected no strong association between endogenous sex hormone levels (totalT, freeT, SHBG) and any KC outcomes (Table 2). Results derived using alternative MR models are shown in Supplementary Tables 6–9 for each exposure, respectively. Estimates derived using only SNP instruments that were more robustly associated ($p < 1 \times 10^{-5}$) with the exposure of interest measured in the independent UKB subset were not meaningfully different (Table 2; Scatter plot in Fig. 2).

MVMR analyses combining MPB and testosterone levels revealed no evidence that the marginal association between MPB and KCs is influenced by endogenous testosterone levels; marginal OR on KC: 1.16 (95% CI 1.06–1.28); BCC: 1.16 (95% CI 1.04–1.29); SCC: 1.30 (95% CI 1.13–1.48) per SD increase in MPB score (Supplementary Table 10).

## Sensitivity analyses to model the potential pleiotropic role of pigmentation-related SNPs

It is possible that some MPB SNPs are pleiotropically associated with skin cancer risk factors. To address this, we first adopted a non-parametric approach to identify potential pleiotropic variants captured by the MR-PRESSO outlier test[28]. We detected four potential SNP-outliers in the association between MPB and KC phenotypes (rs2669871 [near *KRT75*], rs3847069 [near *CUX1*], rs1805007 [non-synonymous functional SNP in *MC1R*], rs12203592 [functional SNP in an enhancer for *IRF4*[29]]); outliers were also confirmed via manual inspection of MR scatter plots (Fig. 3) and funnel plots (Supplementary Figs. 6–9).

PheWAS revealed strong associations between these SNPs and pigmentation traits, including skin colour and ease of skin tanning (Supplementary Table 11; Supplementary Figs. 10–13). Univariable MR analyses support a positive relationship between lighter skin or hair colour and MPB (beta for lighter skin colour on MPB Score=0.28 [95% CI 0.14–0.42] per SD unit increase in MPB; beta for lighter hair colour

**Table 1 | Assessment of instrument strength on major exposures of interest for the two-sample MR analyses**

| Phenotype | unit measurement | Using all variants as SNP instruments | | | Using only variants associated with exposure at $p < 1e-5$ in indep. subset | | Reference |
|---|---|---|---|---|---|---|---|
| | | F-stat | nSNPs | Prop. of variance explained | nSNPs | Prop. of variance explained | |
| Male pattern baldness (MPB) | per SD change in self-reported MPB score | 1800.72 | 473 | 0.140 | 287 | 0.123 | – |
| Total testosterone (totalT) | per SD change in measured endogenous total testosterone | 981.79 | 174 | 0.081 | 117 | 0.074 | Ruth et al. |
| Predicted free testosterone (freeT) | per SD change in derived endogenous free testosterone | 576.95 | 101 | 0.049 | 78 | 0.047 | Ruth et al. |
| Sex hormone-binding globulin (SHBG) | per SD change in measured endogenous SHBG | 1880.57 | 274 | 0.145 | 266 | 0.144 | Ruth et al. |

nSNPs refer to the total number of variants considered in the MR analyses after the harmonisation process. F-stat refers to the F-statistics of the association between the combined SNP instrument and the exposure (phenotype) of interest. See supplementary Data 1 for the individual SNP effect estimates for each exposure of interest.

**Table 2 | Validation of MR association between risk factors and skin cancer risk using SNP instruments derived from independent non-overlapping subset of UKB**

| Cancer outcome | Risk factors | With all SNP instruments identified from entire UKB cohort | | | | | | With only SNP instruments that has p <1e-5 in independent UKB dataset | | | | | |
|---|---|---|---|---|---|---|---|---|---|---|---|---|---|
| | | $P_{UKB}$ | $OR_{UKB}$ | $P_{QSKIN}$ | $OR_{QSKIN}$ | $P_{META}$ | $OR_{META}$ | $P_{UKB}$ | $OR_{UKB}$ | $P_{QSKIN}$ | $OR_{QSKIN}$ | $P_{META}$ | $OR_{META}$ |
| all KC | MPB | 1.45E-04 | 1.17 (1.08 to 1.27) | 0.78 | 1.03 (0.85 to 1.24) | 2.94E-04 | 1.15 (1.06 to 1.23) | 3.86E-03 | 1.16 (1.05 to 1.28) | 0.35 | 1.11 (0.89 to 1.37) | 2.61E-03 | 1.15 (1.05 to 1.26) |
| | TotalT | 0.83 | 1.01 (0.93 to 1.1) | 0.3 | 1.15 (0.88 to 1.5) | 0.61 | 1.02 (0.94 to 1.10) | 0.81 | 1.01 (0.93 to 1.10) | 0.32 | 1.15 (0.87 to 1.52) | 0.6 | 1.02 (0.94 to 1.11) |
| | FreeT | 0.61 | 0.96 (0.83 to 1.12) | 0.89 | 0.97 (0.66 to 1.44) | 0.6 | 0.96 (0.84 to 1.11) | 0.61 | 0.96 (0.81 to 1.13) | 0.74 | 0.94 (0.63 to 1.39) | 0.55 | 0.95 (0.82 to 1.11) |
| | SHBG | 0.22 | 1.05 (0.97 to 1.12) | 0.59 | 1.05 (0.88 to 1.24) | 0.18 | 1.05 (0.98 to 1.12) | 0.22 | 1.05 (0.97 to 1.12) | 0.59 | 1.05 (0.88 to 1.25) | 0.18 | 1.05 (0.98 to 1.12) |
| BCC | MPB | 1.84E-03 | 1.15 (1.05 to 1.26) | 0.49 | 1.1 (0.84 to 1.43) | 1.32E-03 | 1.15 (1.06 to 1.25) | 0.02 | 1.14 (1.02 to 1.28) | 0.31 | 1.17 (0.87 to 1.58) | 0.01 | 1.15 (1.03 to 1.27) |
| | TotalT | 0.37 | 1.04 (0.95 to 1.15) | 0.29 | 1.18 (0.86 to 1.62) | 0.25 | 1.06 (0.96 to 1.15) | 0.45 | 1.04 (0.94 to 1.15) | 0.5 | 1.12 (0.81 to 1.55) | 0.36 | 1.05 (0.95 to 1.15) |
| | FreeT | 0.85 | 0.98 (0.83 to 1.16) | 0.75 | 0.93 (0.57 to 1.5) | 0.78 | 0.98 (0.84 to 1.14) | 0.77 | 0.97 (0.81 to 1.17) | 0.54 | 0.87 (0.55 to 1.37) | 0.62 | 0.96 (0.81 to 1.14) |
| | SHBG | 0.13 | 1.07 (0.98 to 1.16) | 0.39 | 1.11 (0.88 to 1.39) | 0.08 | 1.07 (0.99 to 1.15) | 0.12 | 1.07 (0.98 to 1.16) | 0.35 | 1.11 (0.89 to 1.40) | 0.08 | 1.07 (0.99 to 1.16) |
| SCC | MPB | 2.35E-06 | 1.31 (1.17 to 1.46) | 0.65 | 1.08 (0.77 to 1.53) | 4.25E-06 | 1.28 (1.15 to 1.43) | 1.33E-04 | 1.29 (1.13 to 1.48) | 0.48 | 1.15 (0.77 to 1.72) | 1.19E-04 | 1.28 (1.13 to 1.45) |
| | TotalT | 0.24 | 0.92 (0.81 to 1.05) | 0.31 | 1.24 (0.82 to 1.87) | 0.41 | 0.95 (0.84 to 1.08) | 0.49 | 0.95 (0.83 to 1.09) | 0.42 | 1.19 (0.77 to 1.84) | 0.68 | 0.97 (0.85 to 1.11) |
| | FreeT | 0.46 | 0.92 (0.75 to 1.14) | 0.82 | 1.07 (0.58 to 1.99) | 0.54 | 0.94 (0.77 to 1.15) | 0.59 | 0.94 (0.77 to 1.16) | 0.79 | 1.09 (0.57 to 2.09) | 0.67 | 0.96 (0.78 to 1.17) |
| | SHBG | 0.92 | 0.99 (0.9 to 1.1) | 0.99 | 1.00 (0.74 to 1.36) | 0.93 | 1.00 (0.90 to 1.10) | 0.86 | 0.99 (0.90 to 1.10) | 0.96 | 1.01 (0.74 to 1.37) | 0.88 | 0.99 (0.90 to 1.09) |
| Melanoma | MPB | – | – | – | – | 0.84 | 0.99 (0.91 to 1.08) | – | – | – | – | 0.38 | 0.96 (0.88 to 1.05) |
| | TotalT | – | – | – | – | 0.29 | 0.94 (0.84 to 1.05) | – | – | – | – | 0.34 | 0.94 (0.84 to 1.06) |
| | FreeT | – | – | – | – | 0.91 | 0.99 (0.81 to 1.20) | – | – | – | – | 0.84 | 0.98 (0.78 to 1.22) |
| | SHBG | – | – | – | – | 0.22 | 0.95 (0.88 to 1.03) | – | – | – | – | 0.25 | 0.95 (0.88 to 1.03) |

on MPB = 0.13 [95% CI 0.08–0.18], Supplementary Table 12), suggesting a potential common biological pathway. In our MVMR model incorporating pigmentation variables (hair and skin colour), the marginal association between MPB and KCs (including BCC and SCC separately) showed signs of attenuation towards the null (e.g. OR on KC 1.05 [95% CI 0.98–1.12] adjusted for pigmentation compared with original MVMR OR 1.16 [1.06–1.28]; see Fig. 3). These results suggest that the association between genetic variants associated with MPB and skin cancer body-wide might be driven by a pleiotropic effect on pigmentation. MVMR results for all tested combinations of the five traits (each model satisfying the minimal conditional-F > 10 requirement[30]) are shown in Supplementary Tables 13–15, each revealing similar marginal effect sizes from MPB.

In our second approach, we removed the four pleiotropic SNPs detected via MR-PRESSO from our instrument set and repeated our analyses. Both the revised findings from univariable, and MVMR models, revealed no clear evidence for an overall genetic association between MPB and skin cancers, apart from the association with SCC (MPB-SCC univariate MR estimate: OR 1.17 [95% CI 1.07–1.28], $p = 8.06 \times 10^{-4}$; MVMR estimate: OR 1.15 [95% CI 1.02–1.28], $p = 0.02$), see Figs. 2 and 3. We observed a large reduction in the univariable MR model's Cochran's Q statistics following exclusion of the four pleiotropic SNPs, indicating that the genetic heterogeneity among the SNP effect sizes on KC was much smaller in the revised model (see Supplementary Tables 16 and 17). The revised MR estimates from alternative MR models had largely overlapped 95% C.I. with estimates from the [IVW MR model excluding the SNP-outliers detected by MR-PRESSO] (Supplementary Table 18), indicating minimal evidence of horizontal pleiotropy biases on our findings.

**Stratified MR analysis between MPB and skin cancers by body sites**

**Melanoma.** For cutaneous melanoma, we obtained primary body-site-specific cancer data from the UKB and the Melanoma Institute Australia to investigate evidence for a primary site-specific association between MPB and melanoma. A higher genetic predisposition towards MPB was associated with increased risk of head and neck melanoma (meta-analysed OR 1.31 [95% CI 1.07–1.61] in the original model; OR 1.23 [95% CI 1.00–1.50] in the outlier-robust model). MPB was not associated with cutaneous melanoma at other body sites (Table 3). Similar positive findings were obtained when we evaluate melanoma at the scalp region specifically: estimated OR 1.66 [95% C.I. 0.99–2.80], but with signs of attenuation towards the null in the outlier-robust model estimate (OR 1.33 [0.79 – 2.25]) (Supplementary Table 19). Further splitting the association analysis by evaluating scalp melanoma and head and neck melanoma excluding the scalp region separately yielded very similar findings–indicating that the overall MR association between MPB and head and neck melanoma is primarily driven by melanoma on the scalp region (Supplementary Table 19). We also found no evidence that the MPB-melanoma association differed by Breslow thickness with the 95% confidence intervals on the OR estimates for both thick (OR 1.12 [0.89–1.39]) and thin (OR 1.04 [0.82–1.33]) melanoma largely overlapping (see Supplementary Table 20 for the ORs derived from alternative MR models).

**Keratinocyte cancers.** Combining data for site-specific BCC and SCC separately from the UKB and the QSkin cohort, we found limited evidence for an association between MPB and head/neck BCCs (OR 1.08 [0.99–1.17]) or SCCs (OR 0.92 [0.80–1.05]) using pleiotropy-robust MR models. The estimates for all other body sites were largely consistent with a null effect. MR estimates derived using alternative MR models show consistent effect sizes with the IVW estimate, albeit with much lower precision (Supplementary Table 19).

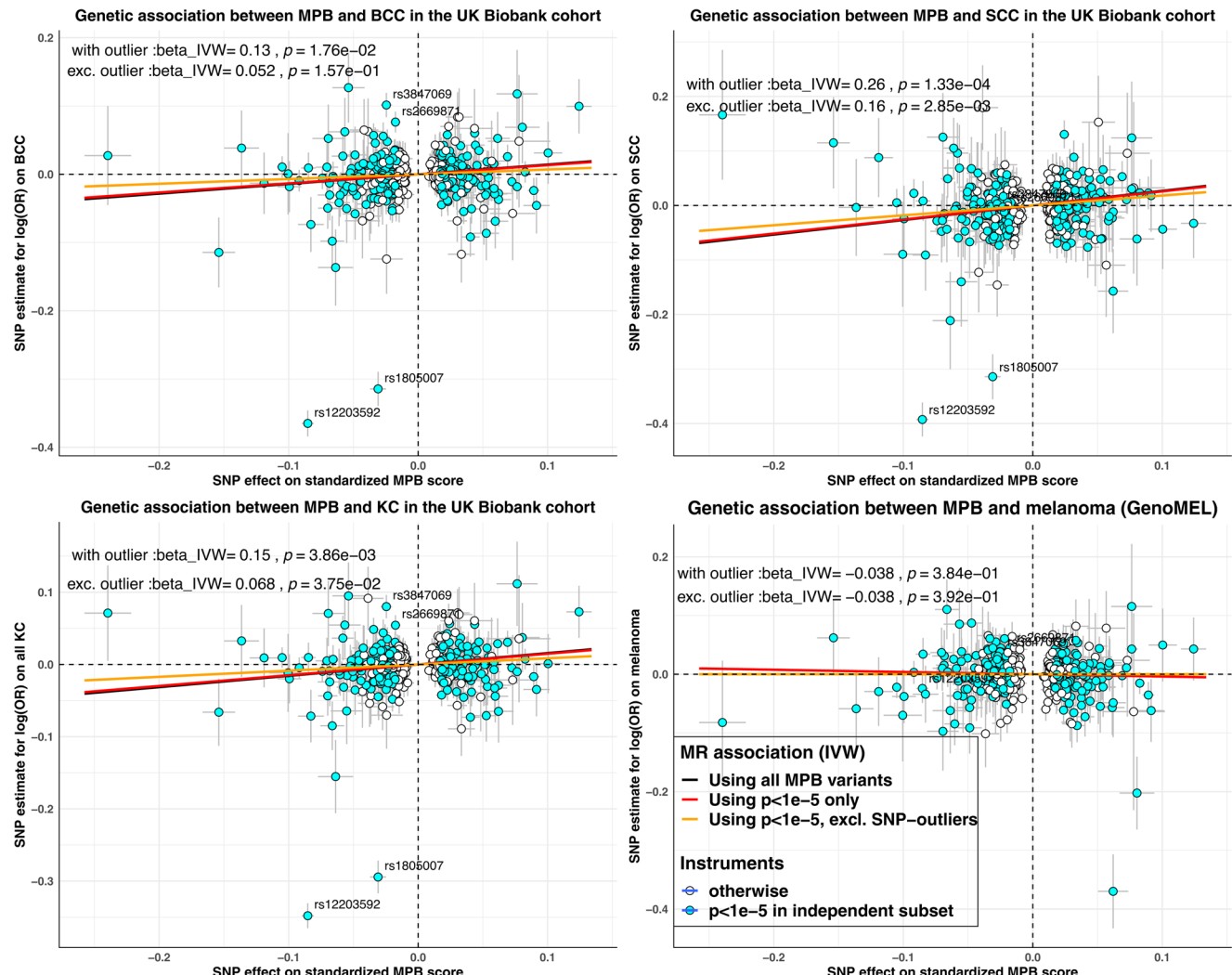

**Fig. 2 | Comparison of MR effect sizes for MPB on skin cancer risk.** beta_IVW refers to the (fixed effect) inverse variance weighted MR effect estimate; p= corresponding two-sided z-test *P*-value of the beta_IVW estimate; MPB Male pattern baldness. Each panel illustrates the MR scatter plot for the association between genetically predicted MPB score and the log(OR) on various types of skin cancer. Each point represents a single MPB SNP instrument, with the corresponding horizontal and vertical error bars reflecting its standard error on the genetic association with MPB and skin cancers, respectively. Points in light blue are MPB variants that showed stronger evidence of association in the UKB subset independent of those used to derive the SNP-skin cancer association. The annotated SNPs on each panel refers to the SNPs identified as outliers via the MR-PRESSO outlier test. Apart from melanoma (right-bottom panel; which shows predominantly null findings), the IVW effect estimates derived using MPB variants after excluding the genetic outliers show strong attenuation of effect sizes towards the null. Source data are provided as a Source Data file.

## Discussion

Melanomas and skin cancers occur with unequal distributions across the body surface, and depending on the country, at higher rates amongst men. While this distribution may be explained by patterns of sun exposure, it has also been postulated that differences in hair covering[5,31], and possibly other hormonal factors[5], may also contribute. We took advantage of several large genetic cohorts that collected site-specific melanoma and skin cancer data, and first showed that testosterone does not play a role on skin cancer susceptibility, a finding supported by our site-specific MR analyses indicating potential increased risk of scalp melanoma among individuals with a high genetic risk for balding. We presented evidence for an association between MPB and risk of SCC and finally revealed insights on mediation mechanisms on the link between MPB and these skin cancers through MVMR.

In Australia, the 2015 age-standardised rates of melanoma per 100,000 were higher for men (63.1) than for women (42.0)[32]. Whilst this trend is also observed in the United States, New Zealand and Canada, the differences were lower or negligible in other countries (e.g. United Kingdom, Denmark and Sweden), and varied by primary body site[5]. Arguments for the involvement of sex hormones in melanoma have primarily stemmed from the observed difference in disease prevalence between men and women, prompting an investigation into potential sex-specific mechanisms that stimulate the proliferation of melanoma cells. Animal studies and in-vivo experiments have suggested that products of testosterone (i.e. progesterone) inhibit melanoma cell growth in a dose-dependent fashion, but there are very few large-scale epidemiological studies exploring this. Of note, our genetic-based approach failed to replicate the association between free testosterone and melanoma susceptibility reported in a recent cross-sectional study[19] using data from the UK Biobank. Here we found no evidence for a causal role between endogenous sex hormone levels (SHBG, freeT, totalT) and skin cancers. Interestingly, our multivariable MR model which enables the estimation of marginal effect sizes for these risk factors on cancer outcomes (upon conditioning on the other competing trait)

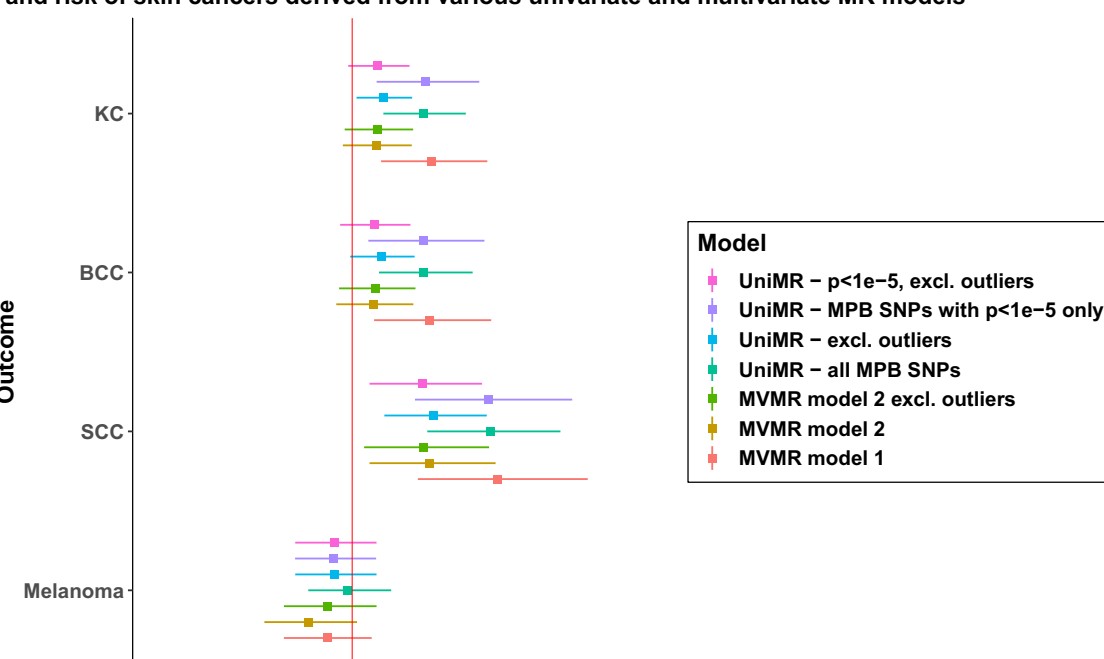

**Comparison for the MR association between genetically predicted MPB and risk of skin cancers derived from various univariate and multivariate MR models**

**Fig. 3 | Comparison of MR-derived association between MPB and various univariable and MVMR models.** UniMR − MPB SNPs with $p < 1e − 5$ only: Univariable MR model for MPB on skin cancer risks using on MPB SNPs with association (z-score) two-tailed P value < 1e-5 on MPB in the independent UK Biobank subset. UniMR − excl. outliers: Univariable MR model for MPB on skin cancer risks using all MPB SNPs excluding pleiotropic SNPs detected by MR-PRESSO. MVMR Model 1: MVMR model incorporating MPB, freeT and totalT. MVMR model 2: MVMR model incorporating MPB, totalT, skin colour and hair colour. The error bars reflect the 95% confidence intervals around each OR estimate. For all MVMR models, the reported OR estimates are the marginal OR estimates for skin cancer per 1 SD increase in MPB upon conditioning on the genetic effect sizes from other traits included in the model. Note that for MVMR model 2, including freeT into the model resulted in the weakening of the combined instrument for MVMR (conditional F-statistics<10 for some traits), which might result in weak instrument bias and hence were not included in the main analysis. However, these findings can be accessed in Supplementary Tables. Source data is provided as a Source Data file.

yielded similar conclusions of a null association between testosterone and skin cancers.

Among the three skin cancers evaluated, SCC showed the strongest association with MPB, followed by BCC. However, we found that the positive association between MPB and risk of KCs was almost completely driven by a functional SNP in the *IRF4* locus (rs12203592). This variant was also previously known to be associated with pigmentation and immune response[29,33–36], thus may influence risk of skin cancers through pathways other than its impact on MPB. Disentangling the observed association between MPB with skin cancers requires careful consideration of both potential causal and pleiotropic mechanisms in play. While the practice of excluding heterogeneous variants and SNP outliers to mitigate potential horizontal pleiotropy bias in the outcomes is a valid approach, we adopted an alternative strategy. Specifically, we chose to model these pleiotropic associations with other traits using MVMR[37]. This decision was motivated by our aim to gain a more comprehensive understanding of potential mediators and genetic confounders that may underlie the relationship between MPB and the development of skin cancers.

In our investigation, we demonstrate how the pleiotropic variants, such as the detected SNP-outlier rs12203592 located in an enhancer for the *IRF4* gene may help explain the observed relationship between MPB and skin cancers (See Fig. 2). For instance, the role that *IRF4* plays in transforming pigmented terminal hair into unpigmented vellus hair is widely established[29,34], suggesting a potential causal role for pigmentation on MPB. rs12203592 has been consistently identified by prior GWAS of nevus count, hair colour, development of freckles and skin pigmentation, all of which are established risk factors for skin cancers[10,29,38,39]. Due to the strong magnitude of association between variants in *IRF4* and both hair loss and pigmentation, it is difficult to determine the type of genetic pleiotropy (vertical or horizontal) exerted by this variant on MPB and skin cancers (i.e. whether the association between *IRF4* on skin cancers were through change in baldness). When pigmentation variables were included in our MVMR model, the association between MPB and KC weakened; though this was also observed when we excluded *IRF4* from the revised univariable MR models. Hence, the association between MPB and overall KC is likely capturing an indirect influence of *IRF4* on both pigmentation-related variables (e.g. skin colour, nevus counts) and risk of balding (with the mode of pleiotropy cannot be reliably determined), through potential pleiotropic effects on autoimmune functions[33] which was not characterised in the present analysis.

As our MR instruments for MPB explain large proportions of phenotypic variances, we had reasonable power to revisit previous observational findings on primary site-specific skin cancer associations with MPB, or at the very least, exclude very large OR effect sizes. Our site-specific MR analysis revealed two interesting key findings. Firstly, the estimated effect size on melanoma located in the head and neck region (based on IVW model excluding SNP outliers) was larger than those of other body sites for both the MIA and the UKB cohorts (Table 3), supporting the role of balding in exposing the scalp area of the skin to UVR radiation. Taken together with our MR findings on testosterone, this demonstrates that testosterone had no direct role in skin cancer formation (i.e. its only role is to remove the natural

**Table 3 | Comparison of estimated MR body-site-specific association between MPB and skin cancer in the UK and Australian population**

| Cancer outcome by anatomical category | Study (UK) | Ncases$_{UK}$ | $P_{UK}$ | OR$_{UK}$ | Study (Aus) | Ncases$_{Aus}$ | $P_{Aus}$ | OR$_{Aus}$ | Fixed-effect meta | |
|---|---|---|---|---|---|---|---|---|---|---|
| | | | | | | | | | $P_{META}$ | OR$_{META}$ |
| *Head and neck region* | | | | | | | | | | |
| Melanoma (all MPB variant) | UKB | 390 | 0.17 | 1.20 (0.92 to 1.57) | MIA | 537 | 0.02 | 1.48 (1.07 to 2.04) | 0.01 | 1.31 (1.07 to 1.61) |
| Melanoma (MPB exc. outlier) | UKB | 390 | 0.24 | 1.18 (0.90 to 1.54) | MIA | 537 | 0.10 | 1.30 (0.95 to 1.76) | 0.05 | 1.23 (1.00 to 1.50) |
| SCC (all MPB variant) | UKB | 1427 | 0.27 | 0.92 (0.80 to 1.07) | QSKIN | 184 | 0.94 | 0.99 (0.65 to 1.49) | 0.29 | 0.93 (0.81 to 1.06) |
| SCC (MPB exc. outlier) | UKB | 1427 | 0.28 | 0.92 (0.80 to 1.07) | QSKIN | 184 | 0.52 | 0.87 (0.57 to 1.32) | 0.22 | 0.92 (0.80 to 1.05) |
| BCC (all MPB variant) | UKB | 4768 | 0.12 | 1.07 (0.98 to 1.16) | QSKIN | 439 | 0.12 | 1.25 (0.94 to 1.67) | 0.05 | 1.08 (1.00 to 1.17) |
| BCC (MPB exc. outlier) | UKB | 4768 | 0.34 | 1.11 (0.90 to 1.37) | QSKIN | 439 | 0.36 | 1.14 (0.86 to 1.51) | 0.08 | 1.08 (0.99 to 1.17) |
| *Upper limb/Arm region* | | | | | | | | | | |
| Melanoma (all MPB variant) | UKB | 446 | 0.01 | 0.72 (0.57 to 0.93) | MIA | 280 | 0.59 | 1.10 (0.78 to 1.55) | 0.08 | 0.84 (0.68 to 1.02) |
| Melanoma (MPB exc. outlier) | UKB | 446 | 0.01 | 0.71 (0.56 to 0.92) | MIA | 280 | 0.72 | 0.94 (0.68 to 1.31) | 0.02 | 0.79 (0.65 to 0.97) |
| SCC (all MPB variant) | UKB | 284 | 0.75 | 1.05 (0.77 to 1.44) | QSKIN | 149 | 0.53 | 1.16 (0.73 to 1.86) | 0.54 | 1.08 (0.84 to 1.41) |
| SCC (MPB exc. outlier) | UKB | 284 | 0.91 | 1.02 (0.74 to 1.40) | QSKIN | 149 | 0.92 | 1.02 (0.64 to 1.63) | 0.88 | 1.02 (0.78 to 1.32) |
| BCC (all MPB variant) | UKB | 636 | 0.47 | 0.93 (0.75 to 1.14) | QSKIN | 248 | 0.49 | 1.15 (0.78 to 1.68) | 0.77 | 0.97 (0.81 to 1.17) |
| BCC (MPB exc. outlier) | UKB | 636 | 0.36 | 0.91 (0.73 to 1.12) | QSKIN | 248 | 0.99 | 1.00 (0.69 to 1.46) | 0.43 | 0.93 (0.77 to 1.12) |
| *Trunk region* | | | | | | | | | | |
| Melanoma (all MPB variant) | UKB | 708 | 0.20 | 0.89 (0.75 to 1.06) | MIA | 1060 | 0.97 | 1.00 (0.78 to 1.30) | 0.30 | 0.93 (0.80 to 1.07) |
| Melanoma (MPB exc. outlier) | UKB | 708 | 0.15 | 0.88 (0.74 to 1.05) | MIA | 1060 | 0.81 | 0.97 (0.76 to 1.24) | 0.19 | 0.91 (0.79 to 1.05) |
| SCC (all MPB variant) | UKB | 209 | 0.74 | 1.05 (0.77 to 1.43) | QSKIN | 23 | 0.98 | 0.99 (0.32 to 3.03) | 0.76 | 1.05 (0.78 to 1.41) |
| SCC (MPB exc. outlier) | UKB | 209 | 0.92 | 1.02 (0.74 to 1.39) | QSKIN | 23 | 0.71 | 0.81 (0.26 to 2.51) | 1.00 | 1.00 (0.74 to 1.35) |
| BCC (all MPB variant) | UKB | 1350 | 0.14 | 1.11 (0.97 to 1.28) | QSKIN | 209 | 0.90 | 0.97 (0.66 to 1.45) | 0.18 | 1.10 (0.96 to 1.25) |
| BCC (MPB exc. outlier) | UKB | 1350 | 0.21 | 1.10 (0.95 to 1.26) | QSKIN | 209 | 0.73 | 0.93 (0.63 to 1.39) | 0.28 | 1.08 (0.94 to 1.23) |
| *Lower limb region* | | | | | | | | | | |
| Melanoma (all MPB variant) | UKB | 239 | 0.27 | 0.84 (0.62 to 1.14) | MIA | 361 | 0.55 | 0.90 (0.65 to 1.26) | 0.22 | 0.87 (0.69 to 1.09) |
| Melanoma (MPB exc. outlier) | UKB | 239 | 0.33 | 0.86 (0.63 to 1.17) | MIA | 361 | 0.55 | 0.93 (0.72 to 1.19) | 0.17 | 0.85 (0.68 to 1.07) |
| SCC (all MPB variant) | UKB | 109 | 0.60 | 0.88 (0.56 to 1.39) | QSKIN | 102 | 0.48 | 1.24 (0.68 to 2.26) | 1.00 | 1.00 (0.70 to 1.43) |
| SCC (MPB exc. outlier) | UKB | 109 | 0.66 | 0.90 (0.57 to 1.42) | QSKIN | 102 | 0.93 | 0.98 (0.55 to 1.75) | 0.69 | 0.93 (0.65 to 1.33) |
| BCC (all MPB variant) | UKB | 269 | 0.85 | 1.03 (0.76 to 1.39) | QSKIN | 93 | 0.63 | 0.87 (0.49 to 1.54) | 0.95 | 0.99 (0.76 to 1.30) |
| BCC (MPB exc. outlier) | UKB | 269 | 0.90 | 1.02 (0.75 to 1.39) | QSKIN | 93 | 0.27 | 0.72 (0.40 to 1.29) | 0.69 | 0.95 (0.72 to 1.24) |

For the MR analyses, 96620, 3176 and 1957 controls were used for the UKB, MIA and QSKIN analyses respectively. P-values here were derived from two-tailed z-tests on the MR association estimate (in log(OR)), unadjusted for multiple testing. SNP outliers (for the repeated analyses excluding potentially pleiotropic/heterogenous SNPs) were determined using the MR-PRESSO outlier test at Bonferroni-corrected P value of 0.05. The model MPB exc. outlier refers to the model utilizing MPB variants excluding the four SNP outliers identified by MR-PRESSO.

protection from UV · hair). We observed the same trend for head and neck BCC (albeit with lower precision), where the magnitude of association was slightly higher for the QSkin cohort compared to the UK Biobank study (Table 3), making it unclear whether these MPB-BCC associations varied between populations in low-UVR (UK) and high-UVR (Australia) geographical regions. For melanoma, we found a strong association between MPB and melanoma at the head and neck region; combined with MPB being a trait that is specific towards men, might help explain the higher prevalence of head and neck melanoma in men as previously reported. However, the estimated effect size between balding and melanoma/KC at the head and neck region (predominantly scalp melanoma) in our datasets (e.g. equivalently head and neck melanoma OR 1.72 [1.14–2.59] per 2 SD increase in MPB score) appeared to be much more modest than those previously reported by Li and colleagues (scalp melanoma HR 7.15 [1.29–39.42] for highest balding category vs. none)[13], though our estimates had much greater precision. Another potential explanation may be that observational associations between balding and skin cancers may be amplified by ascertainment and detection bias, where tumours at the scalp or forehead region are more apparent among patients with MPB.

This study has several notable strengths as compared to previous studies of similar nature. Firstly, the proportion of phenotypic variance explained by our genetic instruments were high ($r^2$ estimates 0.04–0.15; Table 1), attributable to the largely heritable nature of both MPB and sex hormone phenotypes. Combined with the very large sample sizes for melanoma and KCs (approximately 2–4 times larger than those previously reported in any MR findings[40–42]), our statistical power to detect even subtle effect sizes (e.g. OR < 1.3 per SD change in exposure trait) were greatly improved. The polygenic basis of both exposures also enables better assessment of horizontal pleiotropy through various alternative MR techniques, ensuring that our main findings were not severely biased by violation of MR assumptions. In addition, the MVMR enables triangulation of a mediation effect for the association between testosterone or other putative biological pathways in our sensitivity analyses and skin cancers. Finally, we used MR to interrogate whether the association between MPB and skin cancers differed by body site, to compare with previous observational findings.

There are also some limitations to our study. The grading of MPB in the UKB was acquired through participant's self-report; hence we cannot exclude the possibility of self-reporting biases arising from negative social stigma[43]. In practice, the misclassification of MPB scores is unlikely to be directly associated with skin cancer outcomes, as that would generally just reduce the statistical power for (genetic) instrument identification. Hence, the technical difference arising from alternative classifications of MPB is not a potential concern in our MR study. The derived MR finding on melanoma using summary statistics from the melanoma GWAS meta-analysis consisted of participants from the UKB (n = 7782 men), making up ~25% of the total number of cases included in the meta-analysis. However, given that the proportion of overlapping samples is less than 30%, any generated bias on the MR estimate is likely limited[44]. Moreover, we repeated our analyses using only subsets of the UKB participants not involved in the skin cancer GWASs and showed that our findings remain broadly unchanged. Our MR findings assume a linear relationship between balding and cancers, and hence might under-estimate the true effect size if the dose-response relationship violates the linearity assumption[45]. Whilst our analysis on the MIA cohort revealed no evidence that the MPB-melanoma association differed by Breslow thickness, we were unable to repeat this analysis in the UK Biobank as information on tumour thicknesses have not been recorded. Our subgroup analyses revealed evidence that this association is driven by melanoma in the scalp. Whilst this supports our inference on increased UV exposure, we did not have the necessary information/sample size to replicate this stratified MR analysis in the UK Biobank and QSkin cohort. Finally, further external validations will also be required to assess whether these genetic instruments can be applied to probe the balding-skin cancer relationship in other non-European ancestries.

Our approach of efficiently incorporating outlier information as candidate traits in our MVMR model is inspired by the MR-TReasure Your eXceptions (MR-TRYX) framework[37]. The key difference here, however, is that we restricted our candidate traits to those involved in pigmentation whilst the MR-TRYX model exhaustively examines all possible risk factors through a generic PheWAS platform. There may be more optimal candidates, such as immunological factors (e.g. eosinophil count, see Supplementary Table 11), which we did not consider in our approach. In practice, we are largely limited by the number of traits we can simultaneously model in an MVMR framework due to difficulty satisfying the conditional F-stat requirement as we increase the number of traits in the model, distribution of trait-specific conditional F-stat in all our tested MVMR models shown in Supplementary Table 5. Finally, our sample sizes in the body site-stratified analyses were limited and were only drawn from regions with two extreme ends of very low (UK) and very high ambient UV radiation (Australia). It remains unclear on whether our findings can be generalised onto other populations. Hence, replicating our site-specific findings on other populations with moderate UV radiation will help ensure our findings are generalisable.

In conclusion, genetic evidence in this study provides minimal support to the androgen-driven hypothesis linking sex hormones to the development of melanoma and keratinocyte cancers. Pigmentation-related factors very likely mediate the genetic relationship between balding and KCs at all body sites, evident through MVMR findings. Finally, we observed a modest body-site specific association between MPB and both the risk of melanoma and KCs involving the head and neck region greater than other body sites, suggesting that balding might increase susceptibility for melanoma around the head and neck region through reduced hair covering, a potential explanation for sex-differences in head and neck melanoma risk between men and women.

## Methods

### Ethical approval and patient consent
The UK biobank study has been formally approved by the UK Biobank Ethics Advisory Committee. The Qskin and AGDS study (used as controls for the MIA case-control GWAS) has been formally approved by the QIMR Berghofer Human Research Ethics Committee. The MIA study was approved by the Sydney Local Health District Ethics Review Committee at the Royal Prince Alfred Hospital in Sydney, Australia, respectively. This research project is approved by the QIMR Berghofer Human Research Ethics committee. The complete list of HREC involved in the individual studies contributing to the melanoma GWAS meta-analysis can be found in Landi et al.[27] (see Supplementary Notes 1). Written informed consent was obtained from all participants from all studies. The authors confirm that patient data has been obtained according to the terms and conditions of the databases where the data was sourced. More specifically, we have obtained specific permission from the MIA investigators to utilise anonymised genetic and patient data from the MIA study for this project.

### Construction of genetic instruments from the UKB
UKB is a large population-based cohort consisting of predominantly middle-aged (at the time of recruitment) white British participants recruited in the United Kingdom. Participants were genotyped (genetic QC and details available elsewhere[25]) and had extensive phenotypic information collected ranging from self-reported diet and lifestyle behaviour to measurements of disease biomarkers, mental health, medical history and cancer diagnosis. GWAS findings for both MPB and endogenous testosterone levels derived from UKB have been previously reported[10–12].

**MPB.** Data for MPB (UKB-field ID 2395) were available for 227,354 male participants. In the UKB, MPB was self-reported and defined using a 4-point scale of increasing hair loss (see Supplementary Notes 2). We rank-transformed the ordinal MPB scales into standardised Z-scores. Analyses were restricted to only male participants of white British ancestry inferred through ancestral principal component (PC) clusters. We performed two GWASs on the rank-transformed MPB score using: i) all 227,354 available participants for instrument discovery; and ii) the subset of only 90,577 participants not overlapping the UKB KC GWASs.

**Sex hormones.** Endogenous serum total testosterone level (totalT) and sex-hormone-binding globulin (SHBG) were measured as part of the 2019 UKB biochemistry data release. Although publicly available GWAS summary statistics for total and free testosterone for UKB participants are available[11], we adopted a similar phenotype definition to those from Ruth et al.[11] for the derivation of testosterone levels (including bioavailable/free-testosterone [freeT]). The derivation for serum freeT using serum totalT, albumin and SHBG in UKB is described in Supplementary Notes 3. Each GWAS on sex hormones was performed in individuals of white British ancestry using BOLT-LMM v2.3, a linear mixed model GWAS framework that accounts for population structure and cryptic relatedness among samples[46]. We fitted recruitment age, genotype array and the first 20 ancestral PCs as standard covariates for all analyses.

**Re-estimation of SNP effect sizes free from sample overlap.** To obtain unbiased MR estimates that minimize sample overlap in the 2-sample setting[44], we repeated GWAS analyses for MPB and sex hormones using only a subset of the UKB participants independent of those used for the KC GWASs (see below, also Supplementary Notes 4).

### Sex-stratified GWAS summary statistics for melanoma
**All cutaneous melanoma.** Summary statistics from a recent fixed-effects GWAS meta-analysis of clinically confirmed cutaneous melanoma in men only were included[27]. All samples were clinically confirmed cutaneous melanoma. While the majority were invasive melanoma, specific histological subtype data was not available for all sets, and the GWAS meta-analysis will include a small subset of cases with in situ melanomas. Full details describing the participating studies, of analyses, collection of informed consent and ethical approval have been previously reported[27]. Briefly, the following standard GWAS cleaning procedures were performed: exclusion of samples (i) with >3% missingness (ii) with abnormal genotype heterozygosity (iii) was a European ancestry outlier or (iv) related to another sample at identity-by-descent (IBD) PI_HAT > 0.15. SNPs were also filtered where we removed genotypes with a minor allele frequency (MAF) < 0.01 or those with Hardy-Weinberg equilibrium (HWE) $P < 5 \times 10^{-4}$ in controls or $5 \times 10^{-10}$ in cases prior to imputation; our post-imputation analyses were restricted to SNPs with a MAF > 0.005 and imputation quality score > 0.3. A fixed-effects inverse-variance weighted meta-analysis of melanoma risk GWAS was then performed (each GWAS was modelled via logistic regression, including PCs or equivalent control for residual population stratification). The final male-only melanoma risk GWAS included 12,232 cases and 20,566 controls. The distribution of cases across each study is shown in Supplementary Table 1.

**Site-specific primary cutaneous melanoma.** For the site-specific analyses, we obtained site-specific histopathologically confirmed primary cutaneous melanoma diagnoses for men from the UKB and the MIA. In UKB, we derived site-specific primary melanoma diagnoses from ICD-10 codes C43.0-9 (UKB field-ID: 41270). For MIA, melanoma cases with primary site-specific diagnoses and GWAS data were drawn from two complementary data repositories, the MIA Biospecimen Bank (protocol HREC/10/RPAH/530) and Melanoma Research Database (protocol HREC/11/RPAH/444)). Among the 2,236 men diagnosed

with melanoma, more than 99.5% (2227/2236) of the cases were of (or had shown progression into) invasive melanoma. Site-specific primary melanoma cases were then matched against 3,176 melanoma-free controls from the Australian Genetics of Depression Study (AGDS)[47] for the site-specific melanoma GWAS analysis. Details of the definition of anatomical primary site (body sites) categories, data consent and site-specific GWAS analysis in both the UKB and MIA study are available in Supplementary Notes 1, with case distribution across body sites reported in Supplementary Tables 2 and 3. To further investigate whether the association differed by Breslow thickness, we further performed a stratified MR analysis contrasting thick and thin melanoma (see Supplementary Notes 1 under study description on the MIA cohort for the adopted definitions for thick and thin melanoma).

### Sex-stratified GWAS summary statistics for keratinocyte cancers KCs in UK Biobank
We performed sex-stratified GWAS analyses for KC using participants from UKB. Using the case definitions as per previous work[48], we identified 3,483 invasive SCC and 10,718 invasive BCC cases confirmed through ICD-10 and histology records (UKB Field-ID: 40006, 40013) among UKB male participants ascertained through linkage of participant health records to national cancer registries. Men with no prior history of KC or any other cancer diagnosis (n = 96,620) were used as controls. Our analyses were restricted to individuals of white British ancestry identified through self-report and clustering approaches on ancestral principal components[49]. We performed the GWAS analyses for BCC and SCC using SAIGE[50], a recently developed software that implements linear mixed models for binary traits/outcomes, accounting for case-control imbalance and cryptic relatedness. We also performed a GWAS of a combined KC phenotype (cases=13,463; controls=96,620) to leverage the shared genetic architecture between SCC and BCC for improved power[48].

**KCs in QSkin.** In total, 4,049 men with post-quality-control genetic data were clinically diagnosed with KC, with 1,064 and 502 cases identified to have invasive BCC and invasive SCC based on pathological records, respectively. Healthy controls were selected from participants screened/self-reported to have no history of KC or actinic keratoses at the time of recruitment[26]. The cleaning and quality control of the genetic data for the QSkin cohort have been previously described[27]. We performed a sex-stratified GWAS for KC, BCC and SCC, including only male participants of white European ancestry using SAIGE[50], adjusting for recruitment age and the first ten ancestral principal components.

**Site-specific KCs.** Additional body-site-specific GWASs for KCs in men were conducted in the UKB and QSkin cohort. Similarly, these analyses were restricted to those of white European ancestry. For UKB, body-site-specific diagnoses of BCCs and SCCs were obtained through ICD-10 codes C44.0-7. For the QSkin cohort, site data of the KCs were manually recovered through histopathological records. To enable comparison between the two cohorts, we collapsed the site-specific cancers into 4 broad categories (similar to those used for the melanoma analysis): head and neck, upper limb, lower limb and trunk region. Details of the defined categories and the number of cases and controls included for each body site are also presented in Supplementary Notes 5 and Supplementary Tables 2–4.

### Univariable Mendelian randomisation analysis
Prior to the MR analysis, we harmonised the exposures [totalT, freeT, SHBG, MPB] and outcomes (skin cancers) datasets to align alleles and discard palindromic SNPs with non-strand-inferrable allele frequencies (MAF > 0.3). Multi-allelic variants and variants with inappropriate standard errors (i.e. >1 decimal place smaller than those approximated via sample size and minor allele frequency[51], for outcome datasets) were also excluded.

MPB SNP instruments were clumped to remove variants in linkage-disequilibrium (LD) (window = 10 megabase [Mb], max LD $r^2 = 0.001$ in PLINK v1.96b[52]) to ensure strict independence. To control for bias from potentially weak instruments, we additionally curated instrument sets for MPB, totalT, freeT and SHBG to consist only of SNPs that were robustly associated with the corresponding traits at $p < 1\times10^{-5}$ in the independent UKB subset (i.e. no sample overlap with UKB KC GWAS). Statistical power for MR was assessed based on the proportion of variance explained by these instruments (Supplementary Notes 6). The inverse variance-weighted (IVW) estimator was then used to derive the log(odds ratio) (log[OR]) estimates for skin cancers for a standard deviation (SD) increase in the exposure (e.g. genetically predicted endogenous testosterone levels). For SCC, BCC and KC outcomes (and the corresponding site-stratified analyses), we combined the MR log(OR) estimate derived from QSkin and UKB through a fixed-effect inverse variance-weighted model, using the *rmeta* package (v3.0) in the statistical software R.

Alternative MR techniques (MR-PRESSO, MR weighted median, MR weighted mode, MR-Egger, MR-Robust) that relax assumptions regarding horizontal pleiotropy and invalid instruments were also applied to ensure the robustness of our MR inferences[28,53–55]. The specific strength and limitations for each of these methods are reported in the Supplementary Notes 7. All primary MR analyses were performed using the *MendelianRandomization* R package and *TwoSampleMR* R package v0.4.23 curated by the MR-Base platform[56,57]. MR scatter plots were used to inspect statistical outliers, alongside the outlier-test implemented directly via MR-PRESSO[28] (see below).

### Modelling the influence of androgenic and other pleiotropic pathways

We explored the possibility that some MPB-associated SNPs might exert pleiotropic effects that modify the risk of developing skin cancers. However, excluding variants with potentially large effect sizes on the outcome based on (horizontal) pleiotropy is less efficient if information linking the outlier and other pleiotropic pathways can be incorporated via MR. This allows us to understand potential mediators and/or genetic confounders of the exposure-outcome relationship in MR[37]. Here we outline two complementary approaches; full details of these models are in Supplementary Notes 5. We first applied the MR-PRESSO[28] outlier test on MPB vs. skin cancers to identify potential outliers that can bias the MR-IVW results. We then performed phenome-wide association studies (PheWAS) to evaluate the association between the set of potentially pleiotropic SNPs and a series of pigmentation/skin-related phenotypes likely to confound any associations between MPB and skin cancer (see Supplementary Notes 8). This was done by querying the candidate SNPs against two GWAS databases, MRC-IEU OpenGWAS and the OpenTarget platform[56,58,59]. Based on the potential candidate SNP-trait association, we then assessed putative mediation mechanisms driving the association between MPB and skin cancer via the candidate trait. We first attempted to validate the relationship in a univariable MR framework, followed by a multivariable MR (MVMR) analysis incorporating the candidate trait alongside testosterone and MPB on skin cancer (See Fig. 1).

In our second approach, we manually excluded the set of pleiotropic variants identified in the MR-PRESSO outlier test altogether and repeated the univariable and MVMR analyses. Curation of the genetic instruments for the MVMR analysis and the selection criteria for traits to achieve strong instrument strength for the analysis are described in Supplementary Notes 9. The MVMR association analyses were performed using the mv_multiple() function from the *TwoSampleMR* R package[24,56].

### Reporting summary

Further information on research design is available in the Nature Portfolio Reporting Summary linked to this article.

## Data availability

The genetic summary data for the risk factors presented in this manuscript are primarily derived from the UK Biobank cohort (Supplementary Data 1). The UK Biobank phenotype and genetic data can be obtained by application directly to the UK Biobank. The sex-specific melanoma GWAS meta-analysis summary statistics can be obtained through direct application to the study principal investigators (M. H. Law matthew.law@qimrberhofer.edu.au; M. M. Iles M.M.Iles@leeds.ac.uk; M. T. Landi landim@mail.nih.gov). The complete GWAS summary statistics data for KC derived from the QSkin study is available under restricted access, and the data access can be obtained by contacting the following QSkin principal investigators (D. C. Whiteman David.Whiteman@qimrberghofer.edu.au; C. M. Olsen Catherine.Olsen@qimrberghofer.edu.au). The GWAS findings from the site-stratified melanoma data in the MIA is available under restricted access, and data access can be obtained by contacting the MIA investigator R.A. Scolyer (EAscolyer@melanoma.org.au). The summary statistics for SNPs analysed in this study are also provided in Supplementary Data 1. Source data are provided with this paper.

## Code availability

Results from this work were generated using generic open-source MR software codes/functions implemented in R (such as those offered in the R package *TwoSampleMR* v0.4.22, available here https://github.com/MRCIEU/TwoSampleMR) and *MendelianRandomization v0.5.0* (available here https://github.com/cran/MendelianRandomization), illustrated using the ggplot2 v3.2.1R package. Source data are provided with this paper. No custom software codes are used to generate the reported findings. Our analysis code and a test example can be accessed here https://doi.org/10.5281/zenodo.7988335.

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

## Acknowledgements

This work was conducted using data from the UK Biobank Resource (application number 25331), the melanoma GWAS meta-analysis consortium and the QSkin Sun and Health Study. The study was supported by a program grant (APP1073898) from the Australian National Health and Medical Research Council. SM, RAS and DCW are supported by Fellowships from the Australian National Health and Medical Research Council. MS was supported by the Australian Government Research Training Program (RTP) and the QUT Faculty of Health Scholarship. MMI was supported in part by the National Institute for Health and Care Research (NIHR) Leeds Biomedical Research Centre. The views expressed are those of the author(s) and not necessarily those of the NHS, the NIHR or the Department of Health and Social Care. The complete study acknowledgement and grant support for the use of the melanoma GWAS meta-analysis data can be found in Landi et al.[27]. For support on study illustrations, Fig. 1 and Supplementary Fig. 1 were created using BioRender under the number SO25U9BQAU and YD250OKBWU.

*Melanoma Institute Australia*. The creation, support, and analysis of this cohort would not have been possible without the effort and support of patients, staff, and researchers at Melanoma Institute Australia (MIA). The MIA cohort was supported by funding from MIA, the National Health and Medical Research Council (NHMRC), NSW Health Pathology, Cancer Institute New South Wales and infrastructure grants from Macquarie University and the Australian Cancer Research Foundation. Genotyping was also supported by the Intramural Research Program of the Division of Cancer Epidemiology and Genetics, National Cancer Institute, NIH, and by a Worldwide Cancer Research grant 16–0101. We also recognise support from The Cameron Family through MIA.

## Author contributions

J.S.O. designed the study. J.S.O., M.S., X.H., J.C.D., C.M.O., R.E.N., M.T.L., J.S., M.M.I., M.H.L., R.A.S., G.V.L. involved in data preparation. J.S.O., M.S., M.H.L., M.M.I., M.T.L. and J.C.D. analysed the data. M.H.L., S.M., M.S., Y.W., M.M.I., R.A.S., G.V.L., M.I., R.E.N., C.M.O. and D.C.W. provided critical feedback on study design. YW provided illustrations to support presentation study findings. S.M., D.W. and M.H.L. obtained funding for the study. J.S.O., M.S., S.M. and M.H.L. wrote the first draft of the manuscript. All authors contributed to the final draft of the manuscript. The corresponding authors J.S.O. and M.H.L. attest that all listed authors meet authorship criteria and that no others meeting the criteria have been omitted.

## Competing interests

R.A.S has received fees for professional services from F. Hoffmann-La Roche Ltd, Evaxion, Provectus Biopharmaceuticals Australia, Qbiotics, Novartis, Merck Sharp & Dohme, NeraCare, AMGEN Inc., Bristol-Myers Squibb, Myriad Genetics, GlaxoSmithKline. All other authors declare no conflict of interest.

## Additional information

[1]Population Health Department, QIMR Berghofer Medical Research Institute, Herston, QLD, Australia. [2]School of Biomedical Sciences, Faculty of Health, and Institute of Health and Biomedical Innovation, Queensland University of Technology, Kelvin Grove, QLD, Australia. [3]Program in Genetic Epidemiology and Statistical Genetics, Department of Epidemiology, Harvard T.H. Chan School of Public Health, Boston, MA, USA. [4]Division of Cancer Epidemiology and Genetics, National Cancer Institute, National Institutes of Health, Bethesda, MD, USA. [5]Faculty of Medicine, University of Queensland, Herston, QLD, Australia. [6]Melanoma Institute Australia, The University of Sydney, Sydney, NSW, Australia. [7]Department of Melanoma and Surgical Oncology, Royal Prince Alfred Hospital, Sydney, NSW, Australia. [8]Faculty of Medicine and Health, The University of Sydney, Sydney, NSW, Australia. [9]John Curtin School of Medical Research, Australian National University, Canberra, ACT, Australia. [10]Department of Mental Health & Neuroscience, QIMR Berghofer Medical Research Institute, Herston, QLD, Australia. [11]Tissue Pathology and Diagnostic Oncology, Royal Prince Alfred Hospital & NSW Health Pathology, Sydney, Australia. [12]Charles Perkins Centre, The University of Sydney, Sydney, NSW, Australia. [13]Department of Medical Oncology, Royal North Shore Hospital, St Leonards, NSW, Australia. [14]Leeds Institute of Medical Research & Leeds Institute for Data Analytics, University of Leeds, Leeds, UK. [15]NIHR Leeds Biomedical Research Centre, Leeds Teaching Hospitals NHS Trust, Leeds, UK. ✉e-mail: JueSheng.Ong@qimrberghofer.edu.au; matthew.law@qimrberghofer.edu.au

