## [Peer Review File · Nature Communications]

Reviewer comments

Reviewer #1 (Remarks to the Author):

This is a nicely written manuscript on male pattern baldness and risk of overall/site-specific melanoma, SCC and BCC using a Mendelian randomization approach.

1) Is it possible to conduct subgroup analysis for scalp and non-scalp head & neck skin cancer respectively? This may help respond to authors' explanations to the findings, particularly regarding the potential effect of UV and detection bias.

2) Is it possible to conduct subgroup analysis by Breslow thickness of melanoma? It would be interesting to learn the similarity/differences of findings for thin and thicker melanoma.

3) The potential heterogeneous findings by race/ethnicity needs to be considered, unless the study focused on specific race group such as Whites. Would the instrument variables they used be applicable to specific race/ethnicity group?

4) Please specify in the Methods whether the study only focused on invasive melanoma/SCC instead of melanoma/scc in situ.

Reviewer #2 (Remarks to the Author):

This is a really nicely performed Mendelian randomization study, which shows that male pattern baldness is a causal risk factor of melanoma, particularly in the head and neck area. The authors have performed multivariate MR to show that the effect of MPB on melanoma is unlikely to be occurring due to changes in hormone levels, but rather due to pigmentation.

The study is methodologically sound and I have no changes to suggest.

Response to Reviewer

Original reviewer's query in *italics*, our response in blue, with changes made to the manuscript marked in red; for the coloured version

Reviewer 1

1) Is it possible to conduct subgroup analysis for scalp and non-scalp head & neck skin cancer respectively? This may help respond to authors' explanations to the findings, particularly regarding the potential effect of UV and detection bias.

We agree with the reviewer that a subgroup analysis distinguishing scalp and non-scalp head and neck skin cancers will be informative to help us understand potential detection bias. However, in the cohorts we considered, only the MIA cohort had the required sample sizes and the phenotype quality to perform the subgroup analysis (data from the UKB based on ICD-10 codes do not have the degree of nuance to split scalp and neck skin cancers, while the QSKIN cohort is too small, scalp N ~24, to meaningfully perform a well-powered analysis).

Results from the subgroup analysis in MIA are now incorporated into the revised Supplementary Table 19, see also a snapshot of the addition in Appendix Table A. Our findings reveal that the association between head and neck skin cancer and melanoma risk is largely driven by melanoma in the scalp - consistent with our inference that balding increases UV exposure on the scalp region.

The following texts have been added to the results section of the manuscript:

“..Further splitting the association analysis by evaluating scalp melanoma and Head and Neck melanoma excluding the scalp region separately yielded very similar findings - indicating that the overall MR association between MPB and Head and Neck melanoma is primarily driven by melanoma on the scalp region (Supplementary Table 19).”

And we acknowledged the limitation that this subgroup analysis was not performed on other cohorts, in the discussion:

“Our subgroup analyses reveal evidence that this association is driven by melanoma in the scalp. Whilst this supports our inference on increased UV exposure, we did not have the necessary information/sample size to replicate this stratified MR analysis in the UK Biobank and QSKIN cohort.”

2) Is it possible to conduct subgroup analysis by Breslow thickness of melanoma? It would be interesting to learn the similarity/differences of findings for thin and thicker melanoma.

We thank the reviewer for this suggestion. In the MIA dataset, information on Breslow thickness is available. This information is not available for the UK Biobank. We hence split the entire MIA melanoma set into two categories: thin melanoma (≤ 1 mm primary tumour thickness) and thick melanoma (>1 mm) using the breakpoint defined by the AJCC 8th edition guideline (PMID:29923435) and performed the

subgroup analysis. We found no evidence indicating that the observed MR associations between MPB and melanoma differed by Breslow thickness of the tumour, with widely overlapping 95% confidence intervals between both estimates [e.g. the IVW derived OR for all cutaneous melanoma is 1.07(0.88 - 1.29); thick melanoma OR is 1.12(0.89 - 1.39) and thin melanoma OR is 1.04(0.82 - 1.33) respectively]. See also the newly added Supplementary Table 20 in the revised manuscript (also available as Appendix Table B below). We have added this information to the methods, the results and discussion sections.

In methods, we added:

“..To further investigate whether the association differed by Breslow thickness, we further performed a stratified MR analysis contrasted thick and thin melanoma (see **Supplementary Information** under study description on the MIA cohort for the adopted definitions for thick and thin melanoma).”

The corresponding entry in the Supplementary Info document (under study description for MIA) now provide more details on the definition applied to classify thick and thin melanoma:

“For the stratified analysis on Breslow thickness, we adopted the following criteria based on the AJCC 8th Edition T1 guidelines⁵. In brief, the MIA set was split into those whose first primary melanomas was thin (≤ 1 mm cut off used as is the max thickness) (n=765) and thick (>1 mm) (n=1,440).”

We added the following to the results section:

“We also found no evidence that the MPB-melanoma association differed by Breslow thickness with the 95% confidence intervals on the OR estimates for both thick (OR 1.12 [0.89 – 1.39]) and thin (OR 1.04 [0.82 – 1.33]) melanoma largely overlapping (see Supplementary Table 20 for the ORs derived from alternative MR models).”

And the discussion:

“Similarly, we caution the over-interpretation of findings from our anatomically-stratified MR analysis (especially the MR findings contrasting scalp and non-scalp head and neck melanoma) in the MIA as we were unable to perform these analyses in the other two studies due to the lack of relevant information (UK Biobank) and sample size limitations (QSKIN).”

3.1 The potential heterogeneous findings by race/ethnicity needs to be considered, unless the study focused on specific race group such as Whites.

Our analysis is strictly restricted to individuals of European ancestries, hence it is very unlikely to suffer potential heterogeneity driven by mixture of ancestries in the study. More specifically, the genetic instruments (instrumental variables) for our study were derived from individuals of European ancestries in the UK Biobank, identified and matched through ancestral principal components. All of the skin cancer GWASs analysed in our study (UK Biobank, QSKIN, MIA) were also conducted on participants of European ancestries only. This includes all relevant site-specific skin cancer analyses. We focused on European-derived populations because skin cancer is most common in this ancestry, and the sample size is limited in other ancestries.

In the methods section, reads:

(for Male-pattern baldness) “Analyses were restricted to only male participants of white British ancestry inferred through ancestral principal component (PC) clusters.”

We added the following words for clarity: “Each GWAS on sex hormones was performed **in individuals of white British ancestry** using BOLT-LMM v2.3, a linear mixed model GWAS framework that accounts for population structure and cryptic relatedness among samples”

For the outcome GWASs, it was specified that:

“...Our analyses were restricted to individuals of white British ancestry identified through self-report and clustering approaches on ancestral principal components”

We have also added the following to further clarify the case for site-stratified GWASs:

“Additional body-site specific GWASs for KCs in men were conducted in the UKB and QSKIN cohort. **Similarly, these analyses were restricted to those of white European ancestry.**”

3.2 Would the instrument variables they used be applicable to specific race/ethnicity group?

Our analysis in the present study is strictly restricted to those of European ancestry. Whilst it remains plausible that certain genetic variants influencing immune response, pigmentation and regulation of sex hormones are also present in different ancestries, the literature exploring these topics in non-European ancestries remains limited. Hence, we would caution against applying these genetic instruments across ancestries in the absence of external validation.

In the discussion, we have added the following limitation:

“..further external validations will also be required to assess whether these genetic instruments can be applied to probe the balding-skin cancer relationship in other non-European ancestries.”

“Finally, our sample sizes in the body site-stratified analyses were limited and were only drawn from regions with two extreme ends of very low (UK) and very high ambient UV radiation (Australia). **It remains unclear on whether our findings can be generalised onto other populations.** Hence, replicating our site-specific findings on other populations with moderate UV radiation will help ensure our findings are generalisable.”

4) Please specify in the Methods whether the study only focused on invasive melanoma/SCC instead of melanoma/scc in situ.

In our study, the majority of the identified melanoma/SCC cases were invasive. Please see below for a more detailed explanation for each GWAS.

For the cutaneous melanoma GWAS, all cases were clinically confirmed cutaneous melanoma. While the majority of these cases were invasive melanoma, specific histological and behavior subtype information was not available for all sets/studies, and the resultant phase-2 GWAS meta-analysis included a small subset of cases with in-situ melanomas. The specific description for case definitions and the histological

subtype breakdown for each contributing study to the GWAS meta-analysis is available in Landi 2020 (see PMID:32341527).

Melanoma Institute of Australia dataset (used for site-specific melanoma analysis): The MIA dataset consists of 2236 men diagnosed with melanoma, of which only 9 (<0.5%) of these men have had only one melanoma diagnosis and that melanoma was *in situ* and furthermore had no other information that refuted a diagnosis of *in situ* (e.g. also had a AJCC stage I or higher, or lesion thickness > 0). Removing these individuals made no meaningful difference to our overall MR findings.

In both the QSKIN and UK Biobank, all identified cases for melanoma/BCC/SCCs based on ICD-10 C-neoplasm (C43, C44) definitions were invasive melanomas/BCC/SCCs. Additionally, in the QSKIN cohort, BCC and SCC status was also inferred through Australian Medicare treatment code (incision of BCC/SCC tumours) in the Australian medicare system, which would only pick up invasive BCC/SCC tumours.

We have added the following description in the Methods section to clarify the situation for each of the analysed GWASs:

[for melanoma GWAS meta-analysis] “..All samples were clinically confirmed cutaneous melanoma. While the majority were invasive melanoma, specific histological subtype data was not available for all sets, and the GWAS meta-analysis will include a small subset of cases with *in situ* melanomas. Full details describing the participating studies, analyses, collection of informed consent and ethical approval have been previously reported²⁷.”

[for MIA] “For MIA, melanoma cases with primary site-specific diagnoses and GWAS data were drawn from two complementary data repositories, the MIA Biospecimen Bank (protocol HREC/10/RPAH/530) and Melanoma Research Database (protocol HREC/11/RPAH/444)). Among the 2,236 men diagnosed with melanoma, more than 99.5% (2227/2236) of the cases were of (or shown progression into) invasive melanoma.”

[for UKB] “we identified 3,483 invasive SCC and 10,718 invasive BCC cases confirmed through ICD-10 and histology records (UKB Field-ID: 40006, 40013) among UKB male participants ascertained through linkage of participant health records to national cancer registries.”

[for QSKIN] “KCs in QSKIN. In total, 4,049 men with post-quality-control genetic data were clinically diagnosed with KC, with 1,064 and 502 cases identified to have invasive BCC and invasive SCC based on pathological records, respectively.”

Reviewer 2

This is a really nicely performed Mendelian randomization study, which shows that male pattern baldness is a causal risk factor of melanoma, particularly in the head and neck area. The authors have performed multivariate MR to show that the effect of MPB on melanoma is unlikely to be occurring due to changes in hormone levels, but rather due to pigmentation.

The study is methodologically sound and I have no changes to suggest.

We appreciate and thank you again for your time and commitment to review our work.

Appendix (for peer-review use)

Appendix Table A

Snapshot of the added content in Supplementary Table 19

Estimated (meta-analysed) MR ORs for per 1 SD change increase in genetically predicted MPB score on skin cancer risk stratified by major body site categories

Cancer region (cutaneous melanoma)	MR methods	SNPs	Original model		Outlier-adjusted model	
			Pval	OR	Pval	OR
Head and Neck	MR Egger	443	0.07	1.43 (0.97 to 2.11)	0.24	1.26 (0.86 to 1.85)
	Weighted median	443	1.18E-02	1.52 (1.10 to 2.11)	0.02	1.50 (1.07 to 2.10)
	Inverse variance weighted	443	9.67E-03	1.31 (1.07 to 1.61)	0.05	1.23 (1.00 to 1.50)
	Simple mode	443	0.11	1.91 (0.87 to 4.17)	0.11	1.95 (0.87 to 4.39)
	Weighted mode	443	0.25	1.33 (0.81 to 2.18)	0.36	1.25 (0.78 to 2.00)
Head and Neck (excl. Scalp)	MR Egger	443	0.35	1.37 (0.71 to 2.63)	0.66	1.15 (0.61 to 2.19)
	Weighted median	443	0.20	1.43 (0.83 to 2.45)	0.39	1.29 (0.73 to 2.28)
	Inverse variance weighted	443	0.11	1.32 (0.94 to 1.87)	0.28	1.21 (0.86 to 1.69)
	Simple mode	443	0.52	1.58 (0.40 to 6.30)	0.54	1.55 (0.38 to 6.36)
	Weighted mode	443	0.35	1.43 (0.68 to 3.00)	0.27	1.55 (0.71 to 3.38)
Head and Neck (Scalp only)	MR Egger	443	0.13	1.98 (0.81 to 4.83)	0.52	1.34 (0.54 to 3.30)
	Weighted median	443	0.54	1.28 (0.57 to 2.87)	0.69	1.18 (0.53 to 2.64)
	Inverse variance weighted	443	0.01	1.89 (1.18 to 3.03)	0.06	1.57 (0.98 to 2.54)
	Simple mode	443	0.64	1.58 (0.23 to >10)	0.64	1.58 (0.23 to 10.78)
	Weighted mode	443	0.86	1.11 (0.36 to 3.42)	0.86	1.10 (0.38 to 3.22)

Appendix Table B

Estimated MR ORs for per 1 SD change increase in genetically predicted MPB score on cutaneous melanoma risk stratified by (thick vs thin) Breslow thickness in the MIA dataset

Cancer	MR Method	Original estimate		After removal of SNP-outliers	
		Pvalue	OR	Pvalue	OR
Cutaneous melanoma (all thickness)	MR Egger	0.90	1.02 (0.71 to 1.47)	0.49	0.89 (0.63 to 1.25)
	Weighted median	1.00	1.00 (0.75 to 1.34)	0.92	0.98 (0.74 to 1.32)
	Inverse variance weighted	0.52	1.07 (0.88 to 1.29)	0.88	0.99 (0.82 to 1.18)
	Simple mode	0.76	1.11 (0.56 to 2.19)	0.65	1.17 (0.59 to 2.30)
	Weighted mode	0.92	0.98 (0.64 to 1.49)	0.93	0.98 (0.64 to 1.51)
Cutaneous melanoma (thick)	MR Egger	0.85	1.04 (0.69 to 1.58)	0.46	0.86 (0.58 to 1.27)
	Weighted median	0.99	1.00 (0.71 to 1.41)	0.73	0.94 (0.66 to 1.33)
	Inverse variance weighted	0.33	1.12 (0.89 to 1.39)	0.90	1.01 (0.82 to 1.25)
	Simple mode	0.67	1.19 (0.54 to 2.59)	0.65	1.21 (0.53 to 2.79)
	Weighted mode	0.88	0.96 (0.59 to 1.58)	0.94	0.98 (0.62 to 1.55)
Cutaneous melanoma (thin)	MR Egger	0.82	1.05 (0.66 to 1.67)	0.90	0.97 (0.62 to 1.53)
	Weighted median	0.58	1.12 (0.76 to 1.65)	0.93	1.02 (0.68 to 1.52)
	Inverse variance weighted	0.73	1.04 (0.82 to 1.33)	0.96	0.99 (0.78 to 1.26)
	Simple mode	0.77	0.86 (0.32 to 2.36)	0.81	0.88 (0.31 to 2.49)
	Weighted mode	0.98	1.01 (0.57 to 1.76)	0.83	0.94 (0.53 to 1.66)

(additional) Cosmetic changes to Figure 1 in the revised main text.

Justification: The previous illustration fail to highlight the identified SNP-outliers in univariable MR onto the MVMR analysis on potential confounding/mediating pathways lacked clarity. In the revised and visually enhanced diagram, we added an arrow pointing the SNP-outliers from Panel b to Panel c, to draw attention to this connection. The figure caption remains unchanged.

Original illustration:

Revised illustration (a higher resolution of this file in .pdf have been supplied in the revised submission):

Caption:

Figure 1. Schematic diagram outlining the overall study approach of modelling genetic outliers via MVMR. Each panel (a), (b), (c) and (d) is listed in chronological order of the analysis procedure. (a) Schematic univariable MR diagram. (b) Identification of heterogeneous SNP-outliers in the MR-association through MR-PRESSO and MR scatter plot. (c) Selection of candidate traits for inclusion into MVMR via PheWAS findings on SNP-outliers. (d) Modelling the candidate traits into the MVMR analysis to obtain the marginal effect of MPB on skin cancer risk, by conditioning on endogenous testosterone levels and other candidate traits.

Reviewer comments further

Reviewer #1 (Remarks to the Author):

The authors have satisfactorily addressed my comments in the last round. I have no further questions.

Authors: Point-to-point response to the reviewer:

Reviewer #1 (Remarks to the Author): The authors have satisfactorily addressed my comments in the last round. I have no further questions.

Our Response: We are glad that we have satisfactorily addressed the reviewer's comments